# Identification of Metabonomics Changes in Longissimus Dorsi Muscle of Finishing Pigs Following Heat Stress through LC-MS/MS-Based Metabonomics Method

**DOI:** 10.3390/ani10010129

**Published:** 2020-01-13

**Authors:** Jie Gao, Peige Yang, Yanjun Cui, Qingshi Meng, Yuejin Feng, Yue Hao, Jiru Liu, Xiangshu Piao, Xianhong Gu

**Affiliations:** 1State Key Laboratory of Animal Nutrition, Institute of Animal Sciences, Chinese Academy of Agricultural Sciences, Beijing 100193, China; gaojieludou@126.com (J.G.); hkdyangpg@126.com (P.Y.); mengqingshi@caas.cn (Q.M.); yuejin_feng@163.com (Y.F.); haoyueemail@163.com (Y.H.); luckyjiru@126.com (J.L.); 2Institute of Animal Nutrition, College of Animal Science and Technology, Zhejiang A & F University, Lin’an 311300, China; cuiyanjun@zafu.edu.cn; 3State Key Laboratory of Animal Nutrition, Ministry of Agriculture Feed Industry Centre, China Agricultural University, Beijing 100193, China; piaoxsh@cau.edu.cn

**Keywords:** heat stress, pigs, skeletal muscle, metabolites

## Abstract

**Simple Summary:**

Limited research exists on muscle metabolomics of finishing pigs under heat stress. In this study, nine different metabolites in the longissimus dorsi (LD) muscle of finishing pigs under heat stress were screened and identified. Through quantitative verification, it was concluded that the content of L-carnitine in the LD muscles of the finishing pigs could be significantly decreased due to heat stress, which might be a biomarker for monitoring the animal health status and muscle quality under heat stress.

**Abstract:**

Heat stress (HS) negatively affects meat quality by affecting material and energy metabolism, and exploring the mechanism underlying the muscle response to chronic HS in finishing pigs is important for the global pork industry. This study investigated changes in the metabolic profiles of the longissimus dorsi (LD) muscle of finishing pigs under high temperature using ultra-performance liquid chromatography coupled with electrospray ionization quadrupole time-of-flight mass spectrometry (UPLC-ESI-QTOF-MS) and multivariate data analysis (MDA). Castrated male DLY pigs (Duroc × Landrance × Yorkshire pigs, n = 24) from 8 litters were divided into three treatment groups: constant optimal ambient temperature at 22 °C and ad libitum feeding (CR, n = 8); constant high ambient temperature at 30 °C and ad libitum feeding (HS, n = 8); and constant optimal ambient temperature 22 °C and pair-feeding to the control pigs (PF, n = 8). The metabolic profile data from LD muscle samples were analyzed by MDA and external search engines. Nine differential metabolites (L-carnosine, acetylcholine, inosinic acid, L-carnitine, L-anserine, L-α-glycerylphosphorylcholine, acetylcarnitine, thiamine triphosphate, and adenosine thiamine diphosphate) were involved in antioxidant function, lipid metabolism, and cell signal transduction, which may decrease post mortem meat quality and play important roles in anti-HS. Four metabolites (L-carnosine, acetylcholine, inosinic acid, and L-carnitine) were verified, and it was indicated that the muscle L-carnitine content was significantly lower in HS than in CR (*p* < 0.01). The results show that constant HS affects the metabolites in the LD muscle and leads to coordinated changes in the endogenous antioxidant defense and meat quality of finishing pigs. These metabonomics results provide a basis for researching nutritional strategies to reduce the negative effects of heat stress on livestock and present new insights for further research.

## 1. Introduction

Heat stress is a practical problem for livestock worldwide with unequivocal increases in climate warming and the development of highly intensive animal production. The disadvantages of heat stress in animal husbandry are readily apparent and include compromised growth performance, decreased feed efficiency, altered gene expression, and other abnormal physiological reactions [1,2]. Heat stress can influence muscle metabolism and meat quality in animals reared for food production, which may account for considerable economic losses [3].

When the ambient temperature exceeds the thermoneutral zone (16 to 22 °C for growing-finishing pigs), the pigs are considered to be heat-stressed [4]. Chronic hyperthermia leads to increased oxidative stress in various cells and tissues including skeletal muscle [5,6]. The oxidative reaction induced by heat stress has a considerable influence on meat quality. The intermediates in the oxidation process affect the flavor and meat color. Moreover, oxidation destroys the integrity of membranes, leading to disorders of tissue signal transduction and other functions, reducing the nutritional value of meat and its shelf life [7]. A series of studies has also found that the preslaughter exposure of animals to heat stress could alter muscle metabolism and membrane integrity, resulting in undesirable meat characteristics. Changes in skeletal muscle metabolism or physiology can have broad organismal and metabolic impacts due to its large mass. Thus, muscle metabonomics is used to reflect muscle metabolite changes in pigs subjected to heat stress. However, the metabolic changes in heat-exposed livestock and poultry are intricate and multifactorial, and the regulatory mechanisms remain unclear [7,8,9,10,11].

In addition to muscle quality, oxidative stress has been reported to affect growth performance and intestinal function. Chronic mild heat stress (CHS) drastically depresses pig growth performance and compromises intestinal integrity, function, and metabolism [12]. Studies have shown that feed intake and body weight gain are greatly depressed in pigs exposed to high ambient temperatures [13]. Intestinal material transport, digestive capacity, and postabsorptive metabolism are altered during heat stress, which brings about a series of physiological and metabolic changes [14] and decreases body weight (BW) gain [15]. The high thermal loads as well as reduced feed intake and altered gastrointestinal integrity and function directly affect postabsorptive metabolism. However, in recent years, related research found that HS contributes to the decrease in growth performance, dysfunction of skeletal muscle, and changes of intestinal function by causing oxidative stress, and these negative effects are, in large part, independent of reduced feed intake [16]. Therefore, in this study, a group of pigs was pair-fed to heat stressed pigs to allow for distinguishing between the effects of heat stress, per se, from those caused by depressed feed intake. Metabonomics analysis can offer an unbiased view of changes in metabolism, facilitate the characterization of metabolism at the molecular and cellular levels, and characterize pathological states covering entire metabolic pathways to provide a closer link to these functional physiological responses [17,18,19,20]. 

In this study, we hypothesized that heat stress directly (independent of reduced nutrient intake) affects pig skeletal muscle physiological metabolism. Longissimus dorsi (LD) muscle samples were collected after 21 d of continuous mild heat stress at 30 °C and subjected to UPLC-ESI-QTOF-MS analysis, and multivariate data analysis (MDA) was used for the identification of metabolites. The aim of the present study was to investigate the metabonomics response of LD muscle to chronic HS and to determine whether HS has a direct or indirect effect on skeletal muscle physiological metabolism in finishing pigs. Our results are expected to provide new insights into the molecular mechanisms regulating the effects of chronic HS on livestock skeletal muscle.

## 2. Materials and Methods

### 2.1. Chemicals and Reagent

Authentic standards of L-carnosine, inosinic acid (IMP), acetylcholine (ACh), and L-carnitine were purchased from Sigma-Aldrich. HPLC-grade solvents (acetonitrile, methanol, formic) were purchased from Sigma Aldrich (St. Louis, MO, USA). Purified water was obtained from a Millipore Elix (Millipore, Bedford, MA, USA) system. All other chemicals used for experiments were analytical reagent or HPLC grade of commercial resource.

### 2.2. Animal Treatment and Collection of Samples

Castrated 24 male Duroc × Landrance × Yorkshire (DLY) pigs were selected at a body weight of 79.00 ± 1.50 kg from 8 litters from a pig breeding farm in Beijing, China. Within each litter, the three pigs were allocated to one of three treatments: constant optimal ambient temperature at 22 °C and ad libitum feeding (CR, n = 8); constant high ambient temperature at 30 °C and ad libitum feeding (HS, n = 8); and constant optimal ambient temperature at 22 °C and pair-feeding to the control pigs (PF, n = 8). All pigs were individually single-caged in the environmental control chamber provided by the State Key Laboratory of Animal Nutrition, given free access to water, and fed with a fattening diet (corn-soybean meal) containing 15.73% crude protein and 13.39 MJ/kg digestible energy (Appendix A). The relative humidity was controlled at 55% ± 5%, and a 14 h light and 10 h dark cycle was established.

The temperature in the chamber of the HS group increased from 22 to 28 °C on the first day of the test, and increased by 1 °C per hour; on the second day, it increased from 28 to 29 °C; on the third day, it increased to 30 °C; then, a constant temperature in the chamber was maintained at 30 °C for 3 weeks. The PF group maintained a constant temperature of 22 °C, and the PF pigs were fed the average daily intake of the eight pigs of HS group the previous day. 

Before the experiment, the pigs were allowed to acclimatize to the environmental control chamber at 22 °C for 7 d. After 21 d of the trial period, the pigs were electrically stunned and slaughtered after a fasting period of 12 h (overnight). Five minutes after exsanguination, samples of LD muscle from the same area of the right carcass were excised and snap-frozen in liquid nitrogen. Upon arrival at the laboratory, the tubes were stored at −80 °C until use for the metabolomic study.

The experiment was performed in accordance with the guidelines of Beijing Animal Ethics Committee and received prior approval from the Animal Welfare & Ethics Committee of the Institute of Animal Science, Chinese Academy of Agricultural Sciences. Project identification code is IAS20121125.

### 2.3. UPLC-ESI-QTOF-MS Analysis

Sample processing and analysis was done by LC-MS, where 30 mg of homogenate was mixed with 1 mL methanol–water (50:50, *v*/*v*) for 30 min in an ultrasonic bath (Beat Ultrasonic, Zhangjiagang, China). Cellular debris was removed by centrifugation at 16,000× *g* for 30 min, and the supernatant was filtered through a 0.45 μm oil-based membrane. The water-extracted supernatant was subjected to heat treatment at 80 °C for 15 min in a water bath (model no. 283, Thermo Scientific, Marietta, OH, USA) after centrifugation, then immediately cooled and centrifuged at 6000 g for 20 min at 25 ± 4 °C to remove precipitated proteins. The extracts were stored at −80 °C and subsequently used for the analysis on LC-MS.

The aliquots of 5 µL of each sample were subjected to chromatography on a 50 mm × 2.1 mm ACQUITY 1.7 µm BEH C18 column (Waters Corporation, Milford, MA, USA) at a flow rate of 0.4 mL min^−1^, with the column maintained at 50 °C. The chromatographic system used a binary solvent system delivered as a gradient of solvent A (formic acid solution 0.1%) and solvent B (acetonitrile + 0.1% formic acid). The initial gradient conditions were 95% A and 5% B for 3 min followed by a step curvilinear gradient up to 50% B over the next 7 min. The solvent composition was then held at 100% B for 5 min, resulting in a total cycle time of 15 min per sample.

Mass spectrometry was performed on a Waters Xevo G2 Q-TOF Micro coupled with an ACQUITY UPLC system (Waters Corporation, Milford, MA, USA) operating in positive-ion mode. The desolvation gas flow was set to 650 lh^−1^ at a temperature of 350 °C with the cone gas set to 50 lh^−1^ and the source temperature to 100 °C. The capillary voltage and the cone voltage were set to 3000 and 40 V, respectively. In mass spectrometry scanning, data were recorded in the in centroid mode from 50 to 1200 *m*/*z*.

### 2.4. Data Analysis

After UPLC-ESI-QTOF-MS analysis, the centroided and integrated mass chromatographic data were deconvoluted and processed by MarkerLynx 4.1 mass spectrometry software (Waters Corporation, Milford, MA, USA) to generate multivariate data matrices. For data collection, the parameters were set as follows: retention time ranging from 0 to 15 min, mass ranging from 50 to 1200 Da, and mass tolerance at 0.05 Da. For peak integration, the peak width at 5% of the height was 1 s, the peak-to-peak baseline noise was 100, and the peak intensity threshold was 300. No specific mass or adduct was excluded. For data analysis, a list of the intensities of the detected peaks was generated by using retention time (tR) and mass data (*m*/*z*) pairs as the identifier of each peak. This process was repeated for each run until the final sample.

The ion intensities for each detected peak were normalized against the sum of the peak intensities within that sample by using MarkerLynx. The data matrices were exported into SIMCA-P11.5 (UmetricsAB, Umea, Sweden) for data analysis. Principal components analysis (PCA), partial least squares discrimination analysis (PLS-DA), and orthogonal projections to latent structures discriminant analyses (OPLS-DA) were selected for data mining and pattern recognition after optimization. Unsupervised PCA was used to display the natural separation among the heat stress, pair-fed, and control groups in muscle samples by visual inspection of the score plots. Supervised PLS-DA and OPLS-DA were performed to explore the differences between each pair of groups based on the endogenous metabolites by incorporating known classifications.

Metabolites data in LD muscles of different treatments were statistically analyzed using one-way ANOVA (SAS Systems, Release 9.4; SAS Institute Inc., Cary, NC, USA). We used Duncan’s multiple range tests to analyze significant means (significance at *p* < 0.05 and *p* < 0.01).

MarkerLynx lists of the significantly different metabolite candidates were generated on the basis of the corresponding OPLS S-plot and variable importance in projection (VIP) values. Significantly different metabolites were identified based on the retention time, accurate mass by MS and MS/MS, and elemental composition data by comparing results from various databases such as the Human Metabolome Database (HMDB) (http://www.hmdb.ca), the METLIN Metabolite Database (http://metlin.scripps.edu), and MassBank (http://www.massbank.jp). Those metabolites with authentic standard samples were subjected to LC-MS/MS to confirm the identity of ambiguous compounds.

### 2.5. Quantitative Verification of Muscle L-Carnitine Content

The aliquots (5 µL) of each sample were injected, and LC separation was performed using chromatography with a C18 column (150 × 4.6 mm; Waters Corporation, Milford, MA, USA) at 0.8 mL min^−1^ and 50 °C operated by API 3200 Q-TRAP (Applied Biosystems, Foster City, CA, USA) coupled to an HPLC system (LC20A, Shimadzu Instruments, Kyoto, Japan) operating in positive-ion mode. The mobile phase consisted of water + 0.1% formic acid (A) and acetonitrile + 0.1% formic acid (B). Elution was performed over 25 min: 98–72% A for 0–10 min, 72–0% A for 6 min, 0–98% A for 0.1 min, and 98% A for 8.9 min. A standard curve was generated with different concentrations (5–1000 mM) of pure L-carnitine (C0158, Sigma-Aldrich, St. Louis, MO, USA). The resulting linear regression was *y* = 0.004*x* − 5.15e + 0.004 (r = 0.9906), of which *x* is the L-carnitine concentration, and *y* is the mass spectrum response. The retention times and peak areas were analyzed by API Analyst 1.5 (LC20A, Shimadzu Instruments, Kyoto, Japan). The L-carnitine concentration was calculated from the standard curve.

### 2.6. Determination of Intramuscular Fat Content

Diethyl ether extraction was performed after acid hydrolysis to measure intramuscular fat content, following the method described by GB/T 5009.6-2003.

## 3. Results

### 3.1. MDA of Data Matrices

Metabolic profiling of muscle samples from the CR, HS, and PF groups was performed by UPLC-ESI-QTOF-MS. Some visual differences were noted among the base peak intensity chromatographs of muscle samples from three treatments in ESI-positive mode (Appendix A).

After UPLC-ESI-QTOF-MS analysis of the muscle samples and pretreatment of the raw information by MarkerLynx, PCA coupled with PLS-DA was used to profile the metabolome changes in the negative data matrix. PCA is a well-known nonparametric method of classification, which has been shown to be an effective approach to visualize high-dimensional data by projecting the data points into a low-dimensional space. An overview of the muscle metabolic profiling was initially generated by PCA, and the scores t [1] and t [2] in the PCA score scatter plot were the two most important indices for summarizing the observations in the dataset. As shown in the PCA score scatter plot (Figure 1A), the muscle metabolic profiles detected in ESI-positive modes showed no distinction among the CR, HS, and PF groups, while the data in the three groups were clustered together by the PLS-DA score scatter plot (PLS-DA analysis is shown in Figure 1B). In the PLS-DA diagram, the three groups could be basically distinguished, while under the OPLS-DA pattern recognition method, each pair of groups could be completely distinguished without overlap or crossover.

OPLS-DA was performed on the integrated MS data to identify the metabolites that contributed to the classification of samples and to obtain better discrimination among the three groups in our study. Since three treatments (CR, HS, and PF) were used in this experiment, which may affect the multivariate data analysis and, consequently, the clustering of the groups, we repeated the data evaluation by pairwise comparison: CR vs HS, CR vs PF, and HS vs PF. As shown in the three two-component OPLS-DA score scatter plots (Figure 2), the muscle samples from the three groups were distinctly separated from each other. The calculated goodness of fit (R^2^Y) was 0.94, 0.937, and 0.94, and the goodness of prediction (Q^2^Y) was 0.454, 0.611, and 0.576, respectively.

### 3.2. Differential Metabolites in LD Muscles of Different Treatments

The OPLS-DA S-plots revealed those variables that contributed most to group separation, showing “variable importance in projection” (VIP) values. Higher VIP values of ions indicated that these variables had a major contribution to the separation of each group. In the S-plot (Figure 2), each point represents an ion t_R_-*m*/*z* pair, the X-axis represents the variable contribution, and when the distance of an ion t_R_-*m*/*z* pair point is farther from zero, that ion contributes more to the differences among the three groups. The Y-axis represents variable confidence. When the distance of an ion t_R_-*m*/*z* pair points is farther from zero, the ion has a higher confidence level for the difference between two groups. Thus, the ions that deviated from the cloud of ions on the edge of the S-plot (higher VIP values) were putatively changed metabolites.

A *t*-test was applied to identify the featured peaks that were significantly different among the three groups. For those featured peaks, the *P*-values far less than the significant difference level (*p* < 0.1) were exported for signal selection. By combining the results of OPLS-DA and the ion intensity variation plots for each group, metabolite ions with high VIP values were selected for further investigation.

Accurate mass and mass spectrometric fragmentation patterns were utilized to search the KEGG, PubChem, Metlin, and Human Metabolome Databases. We combined database searching with literature mining to infer the metabolites corresponding to the featured peaks. The inferred metabolites that were indicated by the featured peaks in every pair of groups are shown in Table 1 and Figure 2. Combining the results of OPLS-DA and the ion intensity variation plots for every pair of groups, metabolite ions with high VIP values were individually selected from muscle samples for further investigation. Nine metabolites including L-carnosine, acetylcholine (ACh), inosinic acid (IMP), L-carnitine, L-anserine, L-alpha-glycerylphosphorylcholine (L-α-GPC), acetylcarnitine (ALCAR), thiamine triphosphate (ThTP), and adenosine thiamine diphosphate (AThDP) were found to show significant differences among the three groups.

As shown in Table 1, 8 metabolites were tentatively identified based on accurate molecular mass measurements and empirical molecular formulas in the comparison of the HS and CR groups (IMP, L-carnitine, L-anserine, L-α-GPC, ALCAR, AThDP, L-carnosine, and ThTP). In the comparison of the PF and CR groups, 7 metabolites were tentatively identified (ACh, IMP, L-carnitine, L-α-GPC, ALCAR, AThDP, and ThTP). 5 metabolites were found to be obviously different between the HS and PF groups (Ach, IMP, L-anserine, L-α-GPC and L- carnosine).

### 3.3. Qualitative Validation of Differential Metabolites

Standard samples were used to confirm the identity of ambiguous compounds. Muscle and authentic standards of L-carnosine, IMP, ACh, and L-carnitine samples were subjected to LC-MS/MS. The results showed that their t_R_ and *m*/*z* were identical to those of authentic compounds (Figure 3), which verified their status as altered metabolites. Stick diagrams were used for the identification because no pure authentic compounds were synthesized for L-anserine, L-α-GPC, ALCAR, ThTP, and AThDP when the analysis was carried out (Figure 4).

### 3.4. Quantitative Determination of Muscle Differential Metabolites and IMF

As shown in Figure 5, the L-carnitine content in HS and PF was significantly lower than that in CR (*p* < 0.01). The intramuscular fat (IMF) content in the LD muscle of HS and PF pigs was significantly decreased compared with that in CR (*p* < 0.01; Figure 6), which was in accordance with the muscle L-carnitine content.

### 3.5. Correlation Among Metabolites Content and Meat Quality Index

Table 2 lists the correlation coefficients among metabolite levels in LD muscle and meat quality index measured by our previous studies [21]. ThTP concentration was positively correlated with a* values (at 45 min post-mortem; *p* ≤ 0.05), b* values (at 45 min post-mortem; *p* ≤ 0.01) and b* values (at 24 h post-mortem; *p* ≤ 0.01), and negatively correlated with drip loss (at 48 h post-mortem; *p* ≤ 0.01). There was positive correction between L-α-GPC and L* values (at 24 h post-mortem; *p* ≤ 0.01) and electrical conductivity (at 48 h post-mortem; *p* ≤ 0.05). The ACh was negatively correlated with L* values (at 45 min post-mortem; *p* ≤ 0.05) and L* values (at 24 h post-mortem; *p* ≤ 0.01). The results indicate that ThTP, Ach, and L-α-GPC closely related to meat quality especially meat color post-mortem.

### 3.6. Correlation Among Metabolites Content and Meat Oxidation Index

Heat stress causes oxidative stress which can overproduce reactive oxygen species (ROS). It was reported that the superoxide dismutase (SOD), catalase (CAT), glutathione peroxidase (GPx) and glutathione (GSH) activity levels in the liver and muscle changed under HS [22], which was consistent with our previous studies. Table 3 lists the correlation coefficients among metabolite levels in LD muscle and antioxidant parameters measured by our previous studies [21]. ThTP concentration was positively correlated with SOD (*p* ≤ 0.01). There was a positive correlation between ALCAR and MDA (*p* ≤ 0.01), but a negatively correlation with SOD (*p* ≤ 0.01). The IMP was negatively correlated with SOD (*p* ≤ 0.01). The AThDP was negatively correlated with SOD (*p* ≤ 0.01). The results indicate that ThTP, ALCAR, IMP, and AThDP closely related to antioxidant levels of muscle, especially SOD.

## 4. Discussion

Ultra-performance liquid chromatography (UPLC) coupled to electrospray ionization quadrupole time-of-flight mass spectrometry (ESI-QTOF-MS) not only achieves excellent chromatographic separation by increasing the number of peaks detected but also allows exact measurement of the mass of metabolites with higher sensitivity, accuracy, and precision than conventional HPLC. These characteristics may be useful for efficient subsequent structural elucidation in a metabonomic study [23]. The present study utilized a high-throughput UPLC-ESI-QTOF-MS-based approach for the metabonomic profiling of muscle samples from pigs subjected to mild heat stress or hunger stress. A total of nine differential metabolites were identified, including four verified metabolites (L-carnosine, ACh, IMP, L-carnitine) and five identified metabolites (L-anserine, L-α-GPC, ALCAR, ThTP, AThDP). These metabolites are associated with injury repair function, lipid oxidation and protein oxidation, and neuromodulatory functions, which are closely connected with anti-stress ability and meat quality.

### 4.1. The Effect of Heat Stress on the Contents of Meat Quality-Related Substances

L-carnitine and acetylcarnitine (ALCAR) are both closely related to lipid metabolism. Lipid metabolism affects muscle quality in many ways [24]. Studies have confirmed that L-carnitine and ALCAR can affect the growth, flesh color, and flavor of meat by regulating the proliferation and differentiation of myogenic cells, fatty acid oxidation, myoglobin synthesis, and inosinic acid synthesis [25], and they can also participate in the deacylation/reacylation of membrane phospholipids in the process of membrane repair to maintain the stability of the membrane. L-carnitine plays a key role in lipid metabolism because it is the only factor that can transfer free fat acid (FFA) into mitochondria for their oxidization. ALCAR is an endogenous molecule with an important role in improving the energetic state of the cell. It facilitates the uptake of acyl-coenzyme A (acetyl-CoA) into the mitochondria during fatty acid oxidation [26]. ALCAR participates in energy metabolism and lipid mobilization in vivo. Acylcarnitine enters the mitochondria from the cytoplasm under the action of transposase. The reshaped acetyl-CoA can continuously carry out beta-oxidation, promote lipid metabolism, and affect lipid deposition and fatty acid composition (Figure 7).

Compared with our previous study on proteomic changes of the LD muscle in response to chronic heat stress, most of the proteins are involved in carbohydrate metabolism, myofibrillar and cytoskeleton structure, stress response, antioxidant and detoxification, calcium binding, and cellular apoptosis [27]. Among them, glycogen phosphorylase (PYGM), superoxide dismutase (Cu–Zn) (SOD1), and succinyl-CoA synthetase beta-A chain (SCSLA2) were respectively involved in the glucagon signaling pathway (KEGG ID ssc04922), insulin resistance pathway (KEGG ID ssc04931), peroxisome pathway (KEGG ID ssc04146), and peroxisome proliferator activated receptor (PPAR) signaling pathway (KEGG ID ssc03320). Carnitine *O*-palmitoyltransferase 1 (CPT1), the most important rate-limiting enzyme in the oxidation of fatty acids which can act on L-carnitine and acyl-CoA, participate in the same pathways with the above differentially expressed proteins. Given the above, the correlation and consistency between proteomics and metabolomics in LD muscle of the finishing pigs under heat stress has been confirmed.

The muscle L-carnitine contents of HS and PF pigs are significantly lower than that in CR pigs, which was in accordance with the muscle IMF content, indicating that heat stress results in carcass fat depth reduction and weight loss. The contents of IMP and IMF, together with the degree of lipid oxidation, affect the overall quality of meat. There were no significant differences in L-carnitine levels between HS and PF, which shows that the decrease in L-carnitine might be attributed to feed restriction caused by heat stress.

In general, IMP is thought to have a synergistic effect on taste [28]. IMP contributes mostly to the “umami” taste because of the synergistic effect of MSG (monosodium glutamate) and GMP (disodium guanylate) [29]. IMP is the major nucleotide in muscle, and its degradation results in the formation of ribose, which participates in the Maillard reaction and is essential for the development of desirable flavors [30]. The IMF content and fatty acid (FA) composition of muscle play an important role in meat quality, not only in its tenderness and flavor but also in its nutritional value [31]. IMF represents an important meat quality trait because its content is positively correlated with the texture, tenderness, flavor, and juiciness of cooked meat [32]. In this study, the degradation of IMP in the LD of HS and PF compared with that in CR shows that the applied stress has decreased the meat quality, not only the flavor but also the nutritional value. No other literature about the effect of heat or other stress on animal muscle IMP content was found, and the topic may need further investigation [33]. Furthermore, this result shows that the decrease of IMP in LD muscle under heat stress is probably related to the reduction in feed intake caused by heat stress.

The muscle L-carnosine content of HS pigs is significantly lower than that in CR and PF (Table 1), which indicates that the difference is mainly caused by chronic hyperthermia and not by the reduction in feed intake. L-carnosine participates in oxidation in vivo, especially lipid oxidation. L-carnosine has a significant inhibitory effect on lipid oxidation induced by free radicals and metal ions. Low muscle pH can affect a series of meat quality traits, and L-carnosine functions as an intracellular buffer, which can inhibit the decline of muscle pH allowing lipid oxidation to be inhibited effectively under normal physiological pH conditions to protect meat quality.

In this study, the degradation of L-carnitine, L-carnosine, and ThTP in the LD of HS compared with that in CR may be due to its consumption when the oxidation process increased under heat stress to improve the meat quality and nutritional value. The increase of ALCAR together with the decrease of IMF in the LD of HS and PF may be due to the enhancement of fatty acid oxidation under heat stress, which play a role in improving meat quality. Meanwhile, results revealed that the decrease of L-carnitine and IMP in LD may be changed in the physical process caused by depressed feed intake.

### 4.2. The Effects of Heat Stress on Muscle Metabolites Related to Anti-Stress and Anti-Oxidative Damage Ability

According to the results of our previous study, HS increased the abundance of proteins involved in stress response, disrupted the pro-oxidant/antioxidant balance, and affected meat quality of finishing pigs [27].

As shown in Table 1, the L-carnosine content in LD was decreased in HS compared with that in CR and PF, and ALCAR was increased in HS and PF compared with CR, which indicates that the change of L-carnosine and ALCAR was mainly caused by constant heat stress. Additionally, L-anserine was increased in HS compared with CR but decreased in HS compared with PF, which shows that the increase in L-anserine might be attributed to feed restriction caused by heat stress.

L-anserine is a methylated form of L-carnosine that is present at high levels in the breast skeletal muscle of chickens. Owing to their identical chemical structures, with the exception of L-anserine methylation, L-anserine and L-carnosine have equivalent reported physiological functions [34]. L-carnosine has been shown to affect anti-stress and injury repair processes [35,36,37]. L-carnosine can reduce cell injury caused by heat stress by increasing the expression of heat shock proteins, such as HSP72 and HSP90, and can inhibit apoptosis by inhibiting the expression of NF-κB [38]. L-carnosine is a naturally occurring dipeptide that is synthesized by CARNS1 from β-alanine and L-histidine. Our previous research found downregulation of CARNS1 mRNA expression in the muscle of pigs under heat stress [21], which is in accordance with the present research. In contrast to our data, an increase in L-carnosine levels in response to short-term stress was observed in breast and thigh tissue in broilers in another study [39]. This difference may be due to the differences in experimental animals, stress time, and stress intensity. Another possible explanation is that the L-carnosine content in the muscle of livestock and poultry will increase first when stimulated by external stress, and under prolonged stress, the L-carnosine will be exhausted, so that the L-carnosine content in the muscle will be significantly reduced.

Through elaboration of the function of ALCAR, it was found that the significant changes in ALCAR in LD muscle might be caused by high-temperature stress. ALCAR has been shown to prevent carbon tetrachloride-induced oxidative stress in vital tissues of Wistar rats [40]. ALCAR could attenuate the oxidant injury through inhibition of oxidative damage, mitochondrial dysfunction, and ultimately cell apoptosis [30,41,42]. ALCAR was increased in LD muscle under stress, possibly because it was widely needed to repair stress-induced damage and cell apoptosis at the sampling stage of our experiment. It can be indicated from the results that the increase of L-anserine and ALCAR was probably mainly caused by the reduced feed intake caused by heat stress.

### 4.3. Analysis of the Effect of Heat Stress on Neuromodulatory Function in Muscle Tissue

We found that the rectal temperature of pigs was noticeably elevated in Group HS on days 3, 7, 14 and 21 following high-temperature treatment (*p* < 0.01, Appendix A). There is also considerable evidence suggesting that the regulation of body temperature and fluid equilibrium is controlled by the cholinergic system. ACh is an important neurotransmitter in the central cholinergic system. As a neurotransmitter, ACh participates in hypothalamic thermoregulation to regulate body temperature under heat stress. ACh receptors play an important role in the central nervous system. ACh is a neurotransmitter released from the presynaptic membrane and is hydrolyzed to acetic acid and choline under the action of acetylcholinesterase (AChE). Cai Ying et al. found that heat stress can decrease AChE activity in the whole blood of mice, and AChE activity gradually decreases with increasing temperature. Inhibition of AChE activity prevents the clearance of acetylcholine in the synaptic space over time, causing cholinergic hyperactivity and even disability, as changes occur in the responsiveness and tolerance to external stimuli and injuries [43]. L-α-GPC is a small molecule with good biological activity. It represents a large number of natural water-soluble phospholipid metabolites existing in humans and animals. It participates in intermediate metabolism and plays a vital physiological role. When L-α-GPC in animal tissues is absorbed by the human body, it can reach the synaptic end of choline and increase the synthesis and release of ACh. It can promote the biosynthesis of ACh and phosphatidylcholine in the brain and improve memory [44]. In this study, the muscle L-α-GPC content of HS pigs was higher than that in CR and PF groups, while the muscle ACh content of PF pigs was higher than that of CR and HS groups, which indicate that heat stress might affect the release of excitatory neurotransmitters, thereby changing the excitotoxicity of the hypothalamic–pituitary–adrenal axis (HPA).

Thiamine (vitamin B1) is involved in nerve tissue repair and nerve signal modulation and in maintaining the integrity of the myelin sheath, and it affects nerve signal transduction. Thiamine functions predominantly in its phosphorylated derivatives, mainly thiamine diphosphate (ThDP), thiamine triphosphate (ThTP), and thiamine monophosphate (ThMP), while the main role of its cationic form (T+) is as an antioxidant. These derivatives can interact with each other through various enzymatic systems in vivo. ThTP is believed to have metastatic activating neuronal activity and can phosphorylate modified proteins in some animal tissues. Another form, ThDP, can affect glycometabolism as a coenzyme, which in turn affects lipid metabolism, and signal transduction is influenced by the integrity of the cell membrane, which is closely related to cholesterol synthesis in lipid metabolism [45]. In addition, ThDP can promote the synthesis of ACh, which is an important neurotransmitter, and inhibit the decomposition of ACh by cholinesterase. Herein, when thiamine is deficient, the synthesis of ACh decreases, the decomposition of ACh accelerates due to the increase in cholinesterase activity, and nerve conduction becomes poor, which affects the absorption and deposition of nutrients [46]. The biological significance of AThDP remains ambiguous and calls for more attention and study.

In this study, ACh was increased in PF compared with that in HS and CR, and ThTP was decreased in both HS and PF, while L-α-GPC was increased in HS and decreased in PF. These results can be explained, in part, by the possibility that the consumption of ThTP and the synthesis of L-α-GPC were accelerated at the sampling stage of our experiment, and the increase of L-α-GPC may be a preparation for the biosynthesis of Ach in the next stage. Heat stress might lead to changes in the release of neurotransmitters to regulate body temperature and other related life activities by changing the excitotoxicity of the hypothalamic–pituitary–adrenal axis (HPA). It can be indicated from the results that the increase of L-α-GPC and the decrease of Ach and ThTP were probably mainly caused by feed restriction caused by heat stress.

## 5. Conclusions

In conclusion, by investigating changes in the metabolic profiles of LD muscle of finishing pigs under high-temperature conditions using UPLC-ESI-QTOF-MS in combination with MDA, we found that heat stress leads to a range of changes in metabolites in the LD muscle. In particular, significant changes in metabolites related to injury repair, antioxidant, anti-stress, and lipid metabolism were observed. The results indicate that the meat quality (flesh color, flavors, water holding capacity, etc.) and intrinsic antioxidant system of LD muscle tissues will be downregulated by constant heat stress, mainly because of the destruction of membrane integrity, disorders of neurological function, and the peroxidation of proteins and lipids. Combined with the results of our previous study, there was no significant difference in average daily gain and feed intake between HS and PF groups (Appendix A), which were both significantly lower than that in the CR group [47], we can infer that most of the differential expression of metabolites may be due to the reduction of feed intake caused by heat stress, and these changes may not directly affect the growth performance of pigs. We explored the changes in the metabolic profiles of LD muscle under heat stress to provide a basis for research into nutritional strategies for reducing the negative effects of heat stress on livestock and to develop new insights for further research.

## Figures and Tables

**Figure 1 animals-10-00129-f001:**
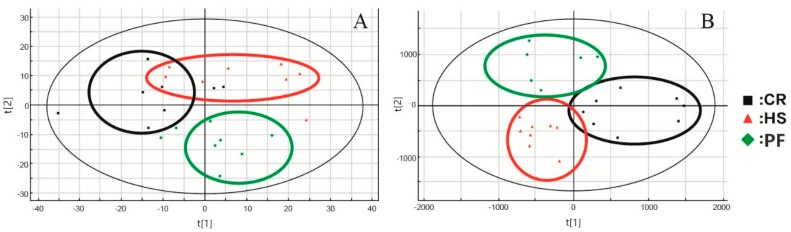
PCA score scatter plots (**A**) and PLS-DA score scatter plots (**B**) based on muscle metabolic profiling of control (CR), heat stress (HS), and feed intake pairing (PF) group pigs in positive ion mode.

**Figure 2 animals-10-00129-f002:**
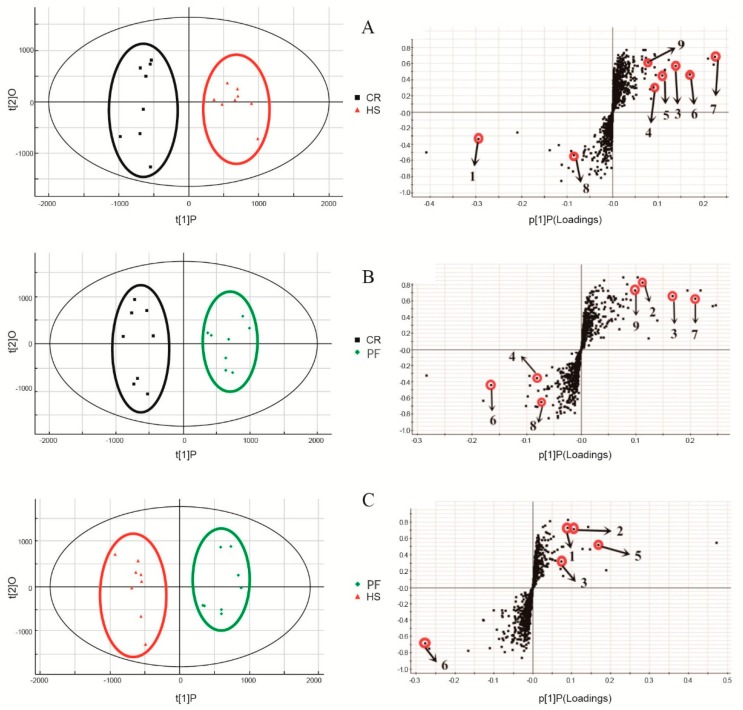
Identification of endogenous metabolites in muscle using UPLC-ESI-QTOF-MS-based metabonomics. The spots marked with different numbers represent different substances that are consistent with the Symbol ID in Table 1. (**A**) OPLS-DA score plots and OPLS S-plot based on muscle metabolites detected in CR and HS pigs in positive ion mode. (**B**) OPLS-DA score plots and OPLS S-plot based on muscle metabolites detected in CR and PF pigs in positive ion mode. (**C**) OPLS-DA score plots and OPLS S-plot based on muscle metabolites detected in HS and PF pigs in positive ion mode.

**Figure 3 animals-10-00129-f003:**
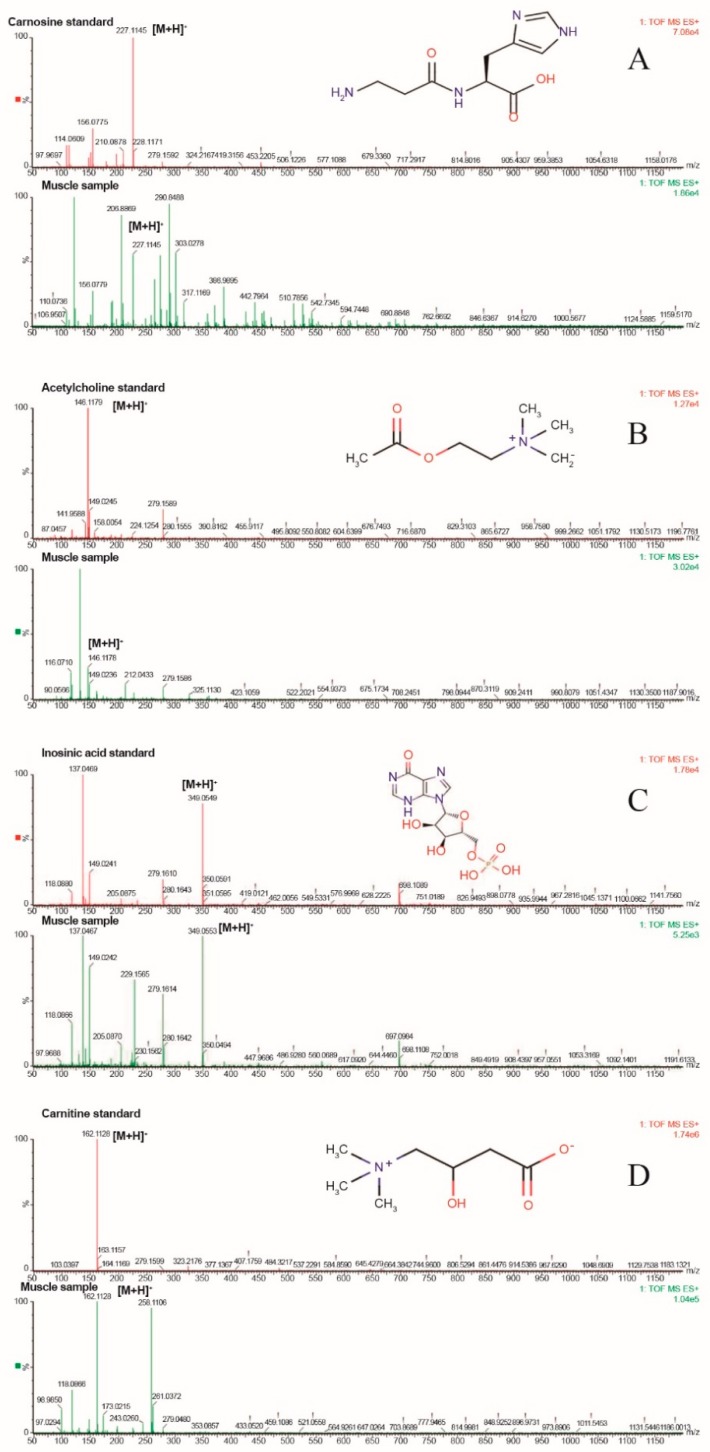
Mass spectra of authentic metabolite standards and muscle samples. Note: The t_R_ and *m*/*z* of the authentic standard of metabolites were identical to the LD muscle sample: (**A**) carnosine, (**B**) acetylcholine, (**C**) inosinic acid, and (**D**) carnitine.

**Figure 4 animals-10-00129-f004:**
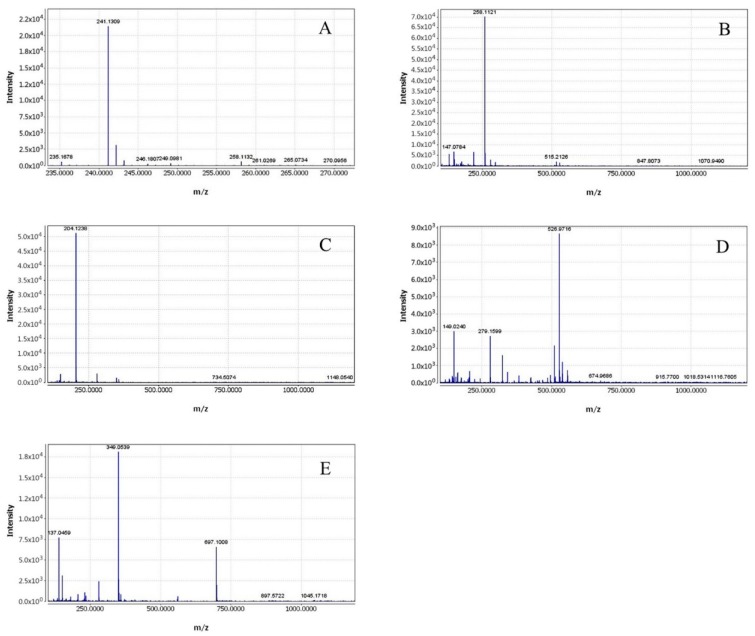
Mass spectra of identified metabolites. (**A**) L-anserine, (**B**) L-α-glycerylphosphorylcholine, (**C**) acetylcarnitine, (**D**) thiamine triphosphate, (**E**) adenosine thiamine diphosphate.

**Figure 5 animals-10-00129-f005:**
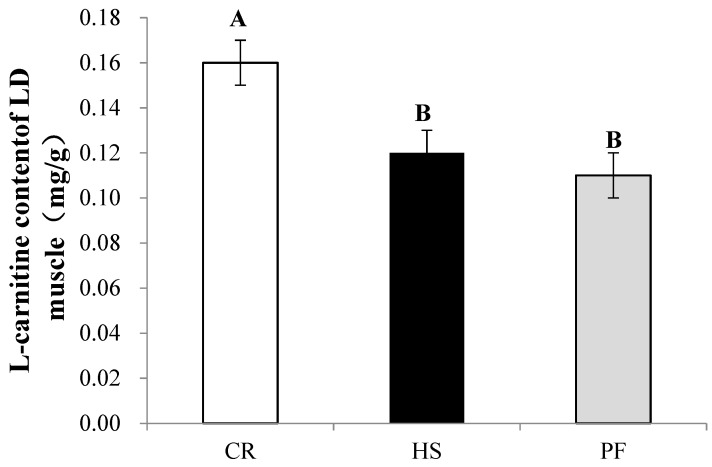
Quantitated analyses of carnitine content of LD muscle in finishing pigs. Values with different capital letter superscripts indicate significant differences (*p* < 0.01), n = 8.

**Figure 6 animals-10-00129-f006:**
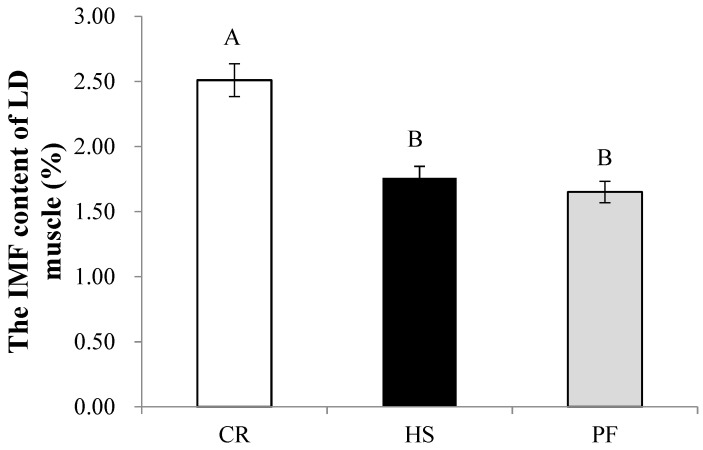
Effects of constant heat stress on the IMF content of LD muscle in finishing pigs. IMF: intramuscular fat in LD muscle, n = 8. Values with different capital letter superscripts indicate significant differences (*p* < 0.01).

**Figure 7 animals-10-00129-f007:**
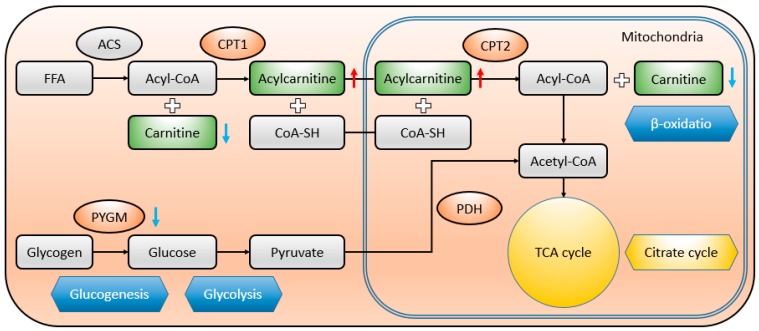
Schematic overview of some important metabolites and major metabolic pathways related to fatty acid (adapted from KEGG ID ssc04931) and carbohydrate metabolism (adapted from KEGG ID ssc04922) in heat-stressed pigs. FFA, free fatty acid, here are mainly long-chain fatty acids (C10-C18). ACS, long-chain acyl-CoA synthetase; CPT1, carnitine O-palmitoyltransferase 1; CPT2, carnitine O-palmitoyltransferase 2; PYGM, muscle glycogen phosphorylase; PDH, pyruvate dehydrogenase, Red up-arrow, heat stress (HS) group vs CR. Blue down-arrow, HS group vs CR.

**Table 1 animals-10-00129-t001:** Differential metabolites in the longissimus dorsi (LD) muscle of the heat stress group (HS, n = 8), feed-intake pairing group (PF, n = 8), and the control group (CR, n = 8) in ESI-positive mode.

Human Metabolome Database ID	Symbol ID	Identity	Formula	t_R_ (min)	HS vs CR			PF vs CR			HS vs PF		
					VIP Values	*p*-Values	Fold Change	VIP Values	*p*-Values	Fold Change	VIP Values	*p*-Values	Fold Change
HMDB00033	1	L-carnosine	C_9_H_14_N_4_O_3_	0.94	11.69	0.002	14.3↓	-	-	-	3.02	0.008	1.45↓
HMDB00895	2	acetylcholine	C_7_H_16_NO_2_	0.89	-	-	-	3.66	0.026	1.47↑	3.25	0.020	1.32↓
HMDB00175	3	inosinic acid	C_10_H_13_N_4_O_8_P	1.52	4.39	0.006	1.58↑	5.70	0.002	1.80↑	2.04	6.94E-5	1.14↓
HMDB00062	4	L-carnitine	C_7_H_15_NO_3_	0.83	2.78	0.031	1.13↓	2.93	0.080	1.13↓	-	-	-
HMDB00194	5	L-anserine	C_10_H_16_N_4_O_3_	0.87	3.10	0.121	1.34↑	-	-	-	5.68	0.073	1.55↓
HMDB00086	6	L-α-glycerylphosphorylcholine	C_8_H_20_NO_6_P	0.79	4.46	0.054	1.20↑	5.05	0.163	1.23↓	9.96	0.014	1.43↑
HMDB00201	7	acetylcarnitine	C_9_H_17_NO_4_	1.32	7.66	0.050	1.20↑	6.89	0.023	1.22↑	-	-	-
HMDB01512	8	thiamin triphosphate	C_12_H_20_N_4_O_10_P_3_S	0.86	2.40	0.010	4.34↓	2.80	0.010	13.4↓	-	-	-
HMDB13647	9	adenosine thiamine diphosphate	C_22_H_30_N_9_O_10_P_2_S	1.51	2.62	0.004	2.12↑	3.44	0.001	2.61↑	-	-	-

Note: t_R_ (min), retention time/minute; VIP values, PLS-DA first principal component variable importance in projection; *p*-values, *t*-test significance; Fold change, ↑ and ↓ indicate that the variable is up-regulated and down-regulated.

**Table 2 animals-10-00129-t002:** Correlation among metabolites content, pH, drip loss, meat color, and electrical conductivity of longissimus dorsi muscle in finishing pigs (Pearson’s linear regression test, n = 24).

Metabolites	pH 45 min	pH 24 h	pH 48 h	Drip Loss 24 h	Drip Loss 48 h	L* 45 min	a* 45 min	b* 45 min	L* 24 h	a* 24 h	b* 24 h	Ec 45 min	Ec 48 h
L-anserine	−0.20	−0.26	0.12	0.20	0.03	−0.18	−0.07	−0.26	−0.26	0.00	−0.12	−0.22	0.35
thiamin triphosphate	0.22	0.27	0.11	−0.30	−0.50 **	−0.24	0.41 *	0.53 **	−0.03	0.08	0.50 **	0.07	−0.23
inosinic acid	−0.21	−0.13	0.22	0.16	0.24	0.04	−0.12	−0.12	−0.09	0.18	−0.00	−0.33	0.17
L-α-glycerylphosphorylcholine	−0.11	0.11	0.10	0.28	0.28	0.40	0.19	0.29	0.59 **	−0.10	0.27	−0.26	0.43 *
L-carnitine	−0.01	0.02	0.22	0.36	−0.07	−0.11	0.22	0.05	0.29	−0.22	0.16	−0.24	0.32
Acetylcarnitine	0.08	−0.15	−0.12	0.06	−0.01	0.08	−0.26	−0.15	0.10	−0.05	0.06	−0.18	−0.24
L-carnosine	0.10	0.35	0.13	−0.10	−0.11	−0.07	0.05	0.34	−0.01	0.20	0.08	−0.16	−0.04
Acetylcholine	0.04	−0.18	−0.06	−0.17	−0.13	−0.39 *	−0.33	−0.35	−0.56 **	0.12	−0.20	−0.09	−0.25
adenosine thiamine diphosphatedenosine	−0.15	−0.13	0.25	0.15	0.23	−0.02	−0.15	−0.19	−0.11	0.15	−0.05	−0.34	0.16

Note: pH, muscle pH; Drip loss; L*, lightness; a*, redness; b*, yellowness; Ec, electrical conductivity; and L-anserine, thiamin triphosphate, inosinic acid, L-α-glycerylphosphorylcholine, L-carnitine, acetylcarnitine, L-carnosine, acetylcholine, and adenosine thiamine diphosphatedenosine are metabolite contents in longissimus dorsi muscle. * 0.01 < *p* ≤ 0.05; ** *p* ≤ 0.01.

**Table 3 animals-10-00129-t003:** Correlation among metabolites content, MDA, SOD, and LDH of longissimus dorsi muscle in finishing pigs (Pearson’s linear regression test, n = 24).

Items	L-anserine	Thiamin Triphosphate	Inosinic Acid	L-α-glycerylphosphorylcholine	L-carnitine	Acetylcarnitine	L-carnosine	Acetylcholine	Adenosine Thiamine Diphosphatedenosine
MDA	0.00	−0.33	0.27	0.31	0.23	0.48 **	−0.35	−0.02	0.28
SOD	−0.39	0.67 **	−0.51 **	0.03	−0.06	−0.48 **	0.32	−0.39	−0.56 **
LDH	0.22	−0.32	0.14	0.01	0.20	0.21	−0.15	0.21	0.20

Note: MDA, malondialdehyde content; SOD, superoxide dismutase activity; LDH, lactate dehydrogenase activity; and L-anserine, thiamin triphosphate, inosinic acid, L-α-glycerylphosphorylcholine, L-carnitine, acetylcarnitine, L-carnosine, acetylcholine, and adenosine thiamine diphosphatedenosine are metabolite contents in longissimus dorsi muscle. ** *p* ≤ 0.01.

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
