# Peer review of "Identification of Metabonomics Changes in Longissimus Dorsi Muscle of Finishing Pigs Following Heat Stress through LC-MS/MS-Based Metabonomics Method"

_animals, 2020, doi:10.3390/ani10010129_

Round 1
Reviewer 1 Report
The manuscript describes a study describing heat stress has negative effects on endogenous antioxidant defense and meat quality by affecting material and energy metabolism in the longissimus dorsi muscle of finishing pigs utilizing UPLC-ESI-QTOF-MS. The manuscript addresses an important issue linking climate change and livestock.
The manuscript is well written and clearly presented.
Questions to the experimental design:
1) Lack of detailed description on temperature control. How the temperature rose to 30℃, directly or gradually?
2) How was the body weight before and after the trial in the CR, PF and HS group?
3) How was the pair-feeding done? One day delay to the heat-stress group?
4) Was the feed intake monitored?
Results and Figures:
Figure 3 - was does the spot number means? Legend does not provide any information.
Questions to writing:
language editing needs further improvement (metabolomic study not metabonomics study…).
Reviewer 2 Report
This is an interesting paper that is examining metabolic changes in the muscle of heat stress, control and pair-fed pigs. Multiple changes in metabolites are reported. The study looks well executed; however, the authors must address the following:
Data needs to be provided showing the heat stress and pair-feeding model worked. More details are needed on how the treatments were administered. The data presented is pointless without pig performance data also being shown. The authors need to discuss the data better in the discussion. They are missing may pig specific heat stress and pair-fed papers from the 2010 to present. The authors have not adequately discussed the pair-fed model results. i.e. what does this data mean in the context of heat stress? Does PF explain half/all the metabolism differences? Pearce et al., 2013 has used the same model in HS. How does fasting influence your results? Line 49: Please mention and cite references for feed intake and intestinal integrity. Line 49: State what these other abnormal reactions are. Line 50: what significant changes? Line 52: Define chronic mild heat stress. Introduction needs to state more specific what the problem is in pig production and heat stress. Metabolomics section can be placed in methods or discussion. Line 77: define LD here. Line 80: Hypothesis? Line 94: single pen not caged, room not cabin. Line 93: The pair-feeding method needs defining. Line 96: Please provide full diet in paper or supplement material. Line 98: room not cabin. Line 99: Please provide the humidity of the rooms. Line 100: change bleeding to exsanguination. Figure 1 not needed. Place in supplementary data. Figure 2: axis labels needed Discussion: Please remove all p-values from discussion. i.e. page 11 line 43-51 reads as results and should be moved there. Likewise A table is needed to show the growth rates, feed intake and body weights of the 21 day study. This information is critical as it defines the phenotype and stress model. Line 1 page 13: “a large numbers of studies” please reference then here. Line 130: no mechanisms explored. Please remove sentence. Line 127: what meat quality aspects? Please define. Line 122: What does this mean for growth? Line 120; did body temperature change? If yes, report it. Please discuss how HS and HS hypophagia impacts muscle metabolism and accretion. This discussion is warranted and should be highlighted in your rationale. How does your data align with the proteomic heat stress pig studies? Should discuss.
